# Nutraceuticals Targeting Generation and Oxidant Activity of Peroxynitrite May Aid Prevention and Control of Parkinson’s Disease

**DOI:** 10.3390/ijms21103624

**Published:** 2020-05-21

**Authors:** Mark F. McCarty, Aaron Lerner

**Affiliations:** 1Catalytic Longevity, San Diego, CA 92109, USA; markfmccarty@gmail.com; 2B. Rappaport School of Medicine, Technion-Israel Institute of Technology, Haifa 3525422, Israel

**Keywords:** nutraceuticals, peroxynitrite, oxidant activity, Parkinson’s disease, prevention, therapy

## Abstract

Parkinson’s disease (PD) is a chronic low-grade inflammatory process in which activated microglia generate cytotoxic factors—most prominently peroxynitrite—which induce the death and dysfunction of neighboring dopaminergic neurons. Dying neurons then release damage-associated molecular pattern proteins such as high mobility group box 1 which act on microglia via a range of receptors to amplify microglial activation. Since peroxynitrite is a key mediator in this process, it is proposed that nutraceutical measures which either suppress microglial production of peroxynitrite, or which promote the scavenging of peroxynitrite-derived oxidants, should have value for the prevention and control of PD. Peroxynitrite production can be quelled by suppressing activation of microglial NADPH oxidase—the source of its precursor superoxide—or by down-regulating the signaling pathways that promote microglial expression of inducible nitric oxide synthase (iNOS). Phycocyanobilin of spirulina, ferulic acid, long-chain omega-3 fatty acids, good vitamin D status, promotion of hydrogen sulfide production with taurine and N-acetylcysteine, caffeine, epigallocatechin-gallate, butyrogenic dietary fiber, and probiotics may have potential for blunting microglial iNOS induction. Scavenging of peroxynitrite-derived radicals may be amplified with supplemental zinc or inosine. Astaxanthin has potential for protecting the mitochondrial respiratory chain from peroxynitrite and environmental mitochondrial toxins. Healthful programs of nutraceutical supplementation may prove to be useful and feasible in the primary prevention or slow progression of pre-existing PD. Since damage to the mitochondria in dopaminergic neurons by environmental toxins is suspected to play a role in triggering the self-sustaining inflammation that drives PD pathogenesis, there is also reason to suspect that plant-based diets of modest protein content, and possibly a corn-rich diet high in spermidine, might provide protection from PD by boosting protective mitophagy and thereby aiding efficient mitochondrial function. Low-protein diets can also promote a more even response to levodopa therapy.

## 1. The Pathogenesis of Parkinson’s Disease—A Key Mediating Role for Peroxynitrite

Within the last decade, a straightforward model for the pathogenesis of Parkinson’s disease (PD) has emerged which appears to be broadly consistent with available evidence. Namely, PD represents a vicious cycle in which activated microglia in the substantia nigra (SN) release cytotoxic factors, most notably peroxynitrite, which damage dopaminergic neurons in a way that induces aggregation and accumulation of alpha-synuclein (ASYN); these ASYN aggregates can promote neuronal death [1,2,3,4,5,6]. Peroxynitrite and excess nitric oxide (NO) also impair the function of dopaminergic neurons by damaging mitochondria, most notably by inhibiting complex I of the mitochondrial respiratory chain [7,8,9]. S-nitrosylation of the E3ubiquitin ligase Parkin and of its binding partner PINK1 can impede the efficiency of protective mitophagy [10,11,12]. The damaged and dying neurons then release certain damage-associated molecular patterns (DAMPs), especially HMGB1 (high mobility group box 1) and aggregated ASYN, which act on microglia to sustain and boost their activation and generation of NO and peroxynitrite—closing the pathogenic circle [13,14]. Studies in rodent and cell culture models of PD confirm that measures which suppress microglial generation of superoxide and/or NO, promote scavenging of peroxynitrite-derived radicals, lessen the expression or modulate the structure of ASYN to lessen its interaction with peroxynitrite, or antagonize the impact of HMGB1 on microglia, quell the damage to dopaminergic neurons [13,15,16,17]. This model explains how certain triggering effects—such as exposure to pesticides which may initiate low-level damage to dopaminergic neurons—can result in a slow smoldering local inflammation that emerges as clinically evident PD once more than half of the dopaminergic neurons in the SN have perished. It also accounts for the fact that PD is more common and emerges earlier in individuals carrying variants of ASYN, which makes them more sensitive to the pro-aggregant effects of peroxynitrite or more toxic to neurons in aggregated form.

This essay proposes that certain nutraceutical strategies—most of them suitable for use by healthy people as measures for promoting overall health—may be useful for preventing or slowing the progression of the pathogenic vicious circle which drives PD. Evidently, such measures would be useful for slowing progression of PD once it has emerged clinically. It should be noted that myeloperoxidase (MPO), though most prominently expressed in neutrophils, can also be expressed in neurons, microglia, and astrocytes, and limited evidence suggests that MPO may contribute to the pathogenesis of PD [18,19,20,21,22]. In particular, chlorination of dopamine by hypochlorous acid, the chief product of MPO, turns it into a potent neurotoxin [23,24]. Since MPO requires hydrogen peroxide as a substrate, suppression of oxidative stress will limit MPO activity.

## 2. Targeting NADPH Oxidase—Phycocyanobilin/Phycocyanin/Spirulina 

Microglial generation of peroxynitrite is contingent on elevated microglial production of superoxide by activated NADPH oxidase (primarily Nox2-dependent), as well as increased induction of the inducible isoform of nitric oxide synthase (iNOS); superoxide and nitric oxide (NO) react spontaneously to yield peroxynitrite, which readily diffuses across cellular membranes and can damage neighboring cells. 

The sustained microglial activation in PD appears likely to reflect DAMP-mediated activation of certain toll-like receptors (notably TLR2 and TLR4), as well as the receptor of advanced glycation end-products (RAGE), the integrin receptor Mac1, and CD11b [25,26,27,28,29,30,31,32,33,34,35]. Rodent and cell culture studies suggest that HMGB1 (which activates TLR4, RAGE, and Mac1), S100B (primarily of astrocyte origin, also a RAGE ligand), and oligomeric ASYN (a ligand for TLR2 and CD11b) contribute to chronic microglial activation in PD [31,34,35,36,37,38,39,40,41,42,43]. S100A8, S100A9, and S100A12 also have the potential to activate RAGE in this disorder; RAGE is highly promiscuous in its response to ligands [14,31,34,35,36,37,38,39,40,41,42,43,44].

Microglial activation of NADPH oxidase is a key feature of PD. How DAMPs promote this activation has not been completely clarified, but HMGB1-mediated activation of Mac1 appears to play a key role [45]. ASYN can also boost microglial NADPH oxidase activity via CD11b [35]. The resulting production of superoxide not only enables peroxynitrite formation, but also, through superoxide’s metabolite hydrogen peroxide, up-regulates signaling pathways that induce iNOS in stimulated microglia [46,47].

With respect to NADPH oxidase, it is known that low nanomolar intracellular concentrations of unconjugated bilirubin generated by heme oxygenase activation function to inhibit certain NADPH oxidase complexes, including those dependent on Nox2 [48,49,50,51,52]. This offers a satisfying explanation for the profound antioxidant activity of heme oxygenase and of the bilirubin it generates; oxidant stress promotes induction of the inducible isoform of heme oxygenase, and the resulting production of bilirubin inhibits a major cellular source of such stress, NADPH oxidase complexes [50,51]. There do not appear to be any prospective studies that have attempted to correlate serum bilirubin with Parkinson’s risk. In case-control studies, results are conflicting as to whether bilirubin is higher or lower in patients; higher bilirubin values might be a marker for increased heme oxygenase-1 activity induced by oxidant stress [53,54,55]. 

Cyanobacteria spirulina, used as a food in certain traditional cultures, and now employed as a nutraceutical supplement, exerts strong antioxidant effects in rodent studies, and is protective in a broad range of rodent models of health disorders—notably those driven by oxidant stress. This has been traced to the fact that spirulina is exceptionally rich in the protein phycocyanin, which carries a covalently attached chromophore—phycocyanobilin (PhyCB)—capable of harvesting light energy which the organism uses to drive ATP production [56,57]. (Hence, it is functionally analogous to chlorophyll). PhyCB is a metabolite of bilirubin’s biosynthetic precursor biliverdin, and, within cells can serve as a substrate for biliverdin reductase; the latter converts it to phycocyanorubin, a compound very close in structure to bilirubin [58,59]. Indeed, there is evidence that phycocyanorubin shares bilirubin’s ability to inhibit NADPH oxidase complexes [59,60]. This offers a satisfying explanation for the ability of orally administered phycocyanin, whole spirulina, or free PhyCB to exert profound antioxidant effects in rodent studies.

Hence, it is not greatly surprising that orally-administered spirulina has been found to be effective in three distinct rodent models of PD [61,62,63,64]. It is reasonable to postulate that a sufficiently high intake of spirulina or of phycocyanin (currently employed as a safe blue food colorant) will retard progression of PD by lessening microglial NADPH oxidase activity. Dose extrapolation from rodent studies suggests that daily intakes of 15–30 g spirulina daily (roughly 1–2 heaping tablespoons, or 100–200 mg PhyCB) may be required to produce the optimally potent antioxidant effects seen in rodent studies with this food [59]. Since most people find that spirulina tastes bad and smells worse, functional foods which mask these characteristics, or nutraceuticals supplying PhyCB-enriched spirulina extracts, may be needed for the practical health-protective potential of spirulina to be realized. 

## 3. Blocking Induction of iNOS—Ferulic Acid, DHA, Vitamin D, Taurine, Cysteine, and EGCG 

The induction of iNOS is a feature of microglial activation and involves joint activation of NF-kappaB and AP-1 transcription factors, each of which bind to the iNOS promoter [65,66,67,68,69,70,71,72]. The sustained microglial activation in PD appears likely to reflect DAMP-mediated activation of certain toll-like receptors (notably TLR2 and TLR4), as well as the receptor of advanced glycation end-products (RAGE) and the integrin receptor Mac1 [73,74,75,76]. Rodent and cell culture studies suggest that HMGB1 (which activates TLR4, RAGE, and Mac1), S100B (primarily of astrocyte origin, also a RAGE ligand), and oligomeric ASYN (a ligand for TLR2 released by dying neurons) contribute to chronic microglial activation in PD [49,52,53,54,55,56,57,58,59,60]. It is notable that RAGE, TLR2, and TLR4 all signal though binding the adapter molecules TIRAP/MyD88; this binding in turn leads to formation of TRAF6-dependent complexes that activate both NF-kappaB and the MAP kinases p38 and JNK [61,62,63,64]. These MAP kinases in turn induce activation of the AP-1 transcription factor required for iNOS transcription; p38 appears to play a particularly critical role in this regard, as its inhibition markedly suppresses iNOS induction [65,69,70,77,78]. The importance of p38 in this regard is partially attributable to the fact that it also acts to prolong the half-life or iNOS mRNA [72,79]. Mac1, which responds to a number of DAMPs, including HMGB1, boosts NADPH oxidase activation via stimulation of phosphatidylinositol-3-kinase; the resulting generation of hydrogen peroxide amplifies signaling pathways leading to iNOS induction [46,47]. Some of the impact of hydrogen peroxide on iNOS induction is mediated by the activation of hypoxia-inducible factor-1, an additional transcription factor which promotes transcription of the iNOS gene [47,80,81]. Hydrogen peroxide might also promote p38/JNK activation via the disinhibition of ASK1, a kinase upstream from these MAP kinases [82,83]. 

The inhibition of NADPH oxidase in microglia or macrophages with either apocynin or bilirubin has been reported to inhibit induction of iNOS driven by TLR4 activation (as with lipopolysaccharide) [46,47]. Catalase overexpression likewise inhibits iNOS induction, consistent with the possibility that hydrogen peroxide-mediated reversible oxidation of cysteine groups plays a role in up-regulating this induction [46,84]. Whether this effect can be generalized to TLR2 and RAGE receptor signaling should be studied. These findings suggest that PhyCB administration might aid PD prevention not only by lessening the availability of superoxide for peroxynitrite generation, but also by down-regulating iNOS induction.

The phytochemical ferulic acid, found in free, but more often in a conjugated form in a wide variety of plant-based foods, may have important nutraceutical potential [85,86]. It is efficiently absorbed and functions potently as a phase 2 inducer, combatting oxidative stress by increasing the expression of a range of antioxidant enzymes, as well as via the induction of the rate-limiting enzyme for glutathione synthase, cysteine-gamma-glutamyl ligase. One of the antioxidant enzymes whose induction ferulic acid promotes is heme oxygenase which, as noted, generates NADPH oxidase-inhibitory bilirubin [87,88,89]. However, ferulic acid also has anti-inflammatory effects not entirely dependent on its antioxidant activity. A recent study has found that ferulic acid inhibits TLR4/MyD88 signaling in pheochromocytoma cells through an effect that is abrogated by MyD88 over-expression [90]. This finding suggests that ferulic acid interferes with MyD88 function—in which case, ferulic acid may have the potential to interfere with the DAMP receptors TLR2, TLR4, and RAGE via interaction with MyD88. This might help to explain why ferulic acid has shown protective activity in both the rotenone and MPTP-induced models of PD in rodents, and to prevent up-regulation of NO production in microglia and astrocytes treated with lipopolysaccharide [91,92,93,94]. The impact of ferulic acid on DAMP-signaling in microglia should receive more study. (A corollary is that, if ferulic acid can interfere with RAGE signaling, it should be useful for controlling diabetic complications.)

Diets enriched in the long-chain omega-3 fatty acids found in fish—and more specifically docosahexaenoic acid (DHA)—have been found to be protective in various rodent models of PD [95,96,97,98,99,100,101,102,103,104,105,106,107]. Decreased expression of inducible NOS, or of nitrite, has been noted in the striatum of PD rodents treated with omega-3s—pointing to suppression of microglia iNOS induction as a likely mechanism for the observed protection [106,107]. These findings may be clinically relevant, as increased dietary unsaturated fats, including long-chain omega-3s, have been associated with decreased PD risk in most but not all pertinent epidemiological studies [108,109]. A possible way that DHA might favorably influence PD risk is via the increased production of the DHA-derived mediators resolvin D1 and resolvin D2, which protect the substantia nigra of rodents when injected intrathecally, and which are protective in cell culture models of PD [110,111,112,113]. These agents activate ALX/FPR2 receptors to promote the synthesis of microRNAs with anti-inflammatory activity [114,115]. In particular, MiR-146b destabilizes mRNAs coding for mediators of TLR4 signaling, including TLR4, MyD88, IRAK-1, and TRAF6—signaling intermediates in iNOS induction [116,117]. Studies with ALX/FPR2 inhibitors will be required to determine whether resolvins mediate the protective effects of DHA in PD by suppressing iNOS induction. An adjunctive possibility is that activation of the microglial GPR120 receptor mediates anti-inflammatory effects of long-chain omega-3s in PD [118,119]. This can intervene in TLR4 signaling at the level of TAK1 via sequestration of beta-arrestin [119,120]. 

A meta-analysis has shown that poor vitamin D status correlates with increased risk for PD, whereas vitamin D supplementation and outdoor work correlate with decreased risk [121]. Good vitamin D status exerts a feedback anti-inflammatory effect on microglia, and the following mechanism may account for this: Microglial activation induces increased expression of the 1-alpha-hydroxylase which converts circulating 25-hydroxyvitamin D to the active hormone calcitriol. This in turn, via the activation of the vitamin D receptor, boosts transcription of the gene coding for MAP kinase phosphatase-1 (MKP-1), a functional antagonist of p38 MAP kinase [122,123]. This increase in MKP-1 inhibits iNOS induction via its suppressive effect on p38 MAP kinase activity. Evidently, this feedback mechanism will be most effective when plasma levels of 25-hydroxyvitamin D are elevated, as they are when dermal production or supplemental intake of vitamin D is high. Hence, maintaining good vitamin D status can be expected to down-regulate NO production by activated microglia.

Another natural agent which inhibits p38 MAP kinase activation in microglia is the physiological gas, hydrogen sulfide (H_2_S). Both endogenously-generated and exogenously-applied H_2_S inhibit the induction of iNOS in activated microglia; this phenomenon is associated with, and mimicked by, the suppression of p38 MAP kinase activation [124,125,126]. The direct molecular target in of H_2_S in this respect has not yet been characterized, but one likely possibility is that H_2_S up-regulates thioredoxin expression and function, thereby down-regulating ASK1 activity [127,128,129]. H_2_S-mediated induction of thioredoxin may represent a homeostatic mechanism, as thioredoxin efficiently reverses the modulatory protein persulfidations induced by H_2_S [130]. When the gene for the chief cerebral source of H_2_S, cystathionine beta-synthase (CBS), is overexpressed in the striatum of mice via stereotaxically-delivered viral vectors, the mice are partially protected from the neurotoxicity and functional impairments induced by MPTP treatment [131]. 

Injectable sodium hydrogen sulfide, as well as a H_2_S-releasing drug, exert an ameliorative effect in rodent PD models, and it has been suggested that the approved drug, sodium thiosulfate (which likewise generates H_2_S), might be useful for management of PD [126,131,132]. With respect to nutraceutical strategies for promoting microglial H_2_S production, both taurine and N-acetylcysteine may have potential. Taurine, whose vascular-protective effects are mediated largely by vascular induction of H_2_S synthesizing enzymes (including CBS), has recently been shown to support brain CBS expression in a rat model of cerebral hemorrhage [133,134]. Cysteine is the limiting substrate for H_2_S synthesis, and its tissue levels can be boosted with supplemental N-acetylcysteine (NAC) [135]. These considerations may rationalize reports that both taurine and NAC are protective in rodent models of PD [136,137,138,139]. Some of the protection afforded by NAC may be mediated by increased synthesis of the important cellular antioxidant glutathione, levels of which are reduced in the SN of PD patients [140,141,142,143]. 

Glucagon-like peptide-1 (GLP-1), via its characteristic receptor, exerts anti-inflammatory effects on microglia and macrophages. In particular, this agent inhibits activation of AP-1 and NF-kappaB in macrophages, thereby suppressing the induction of iNOS [144,145,146,147]. GLP-1 readily crosses the blood–brain barrier, and hence can act on microglia in the CNS; the long-lasting GLP-1 mimetic drug exenatide shares this property. As might be expected, exenatide has shown utility in rodent PD models [148,149,150,151,152,153,154]. While this agent might have some potential in PD therapy, dietary measures which amplify GLP-1 production by the intestinal ileum may have preventive value [155,156,157,158,159]. Notably, short-chain fatty acids such as butyrate produced by bacterial fermentation of dietary fiber in the lower small intestine evoke increased release of GLP-1 into the circulation [160]. Diets rich in fermentable fiber or resistant starch, perhaps used in conjunction with pro-biotics that generate short-chain fatty acids, could be expected to elevate diurnal GLP-1 levels and thereby at least modestly lower PD risk. In mice, direct feeding of sodium butyrate confers protection in PD models [157].

## 4. Caffeinated Beverages also Down-Regulate iNOS Induction 

Stimulation of A2A adenosine receptors by extracellular adenosine can also up-regulate microglial activation and iNOS expression by boosting p38 and ERK1/2 MAP kinase activities [161,162,163,164,165]. This may explain why regular consumption of beverages rich in methylxanthines—notably caffeine, which functions as an antagonist of A1 and A2A adenosine receptors in concentrations achievable by beverage ingestion—has been linked to decreased risk for PD and other neurodegenerative disorders associated with neuroinflammation [166,167,168]. Caffeine can also act to oppose excitotoxicity, as presynaptic A2A adenosine receptors promote glutamate release; excitotoxicity appears likely to contribute to neuronal death in PD [169,170]. The utility of caffeine and other A2A adenosine receptor antagonists in animal models of PD is well documented [171,172]. 

Several studies with cell culture or mouse models of PD have found that the primary polyphenol of green tea, epigallocatechin-gallate (EGCG), confers protection [173,174,175,176,177,178]. In one of these studies, EGCG treatment was shown to block iNOS induction, and the authors postulated that this made a key contribution to the observed benefit [173]. The epidemiology on tea consumption and PD risk is not entirely consistent, and is confounded by the protective effect of caffeine, but a meta-analysis did find that PD risk decreased linearly with increased tea consumption [179,180,181,182,183]. The molecular target of EGCG in regard to PD risk is unclear, but it seems likely to be the 67kDa laminin receptor, stimulation of which by EGCG suppresses TLR4 expression in macrophages and a range of other cell types [184,185,186,187,188,189]. Consumption of the benifuuki strain of green tea—or of polyphenol preparations prepared from it—may be of particular value, as it is rich in a methylated metabolite of EGCG that is far more absorbable than EGCG per se, but is equally as active as an agonist for the 67 kDa laminin receptor [166,179,180,181,182,183,190,191,192]. 

## 5. Scavenging Peroxynitrite-Derived Radicals—Benefits of Supplemental Zinc and Inosine

Efficient scavenging of peroxynitrite-derived radicals within dopaminergic neurons has the potential to suppress tyrosine nitration of ASYN and other neurotoxic effects of this potent oxidant. The cysteine-rich protein metallothionein (MT) has been shown to function physiologically as a peroxynitrite scavenger [193,194,195,196]. Dopaminergic neurons over-expressing MT are protected from peroxynitrite-generating agents [197]. Zinc supplementation can boost MT expression throughout the body, via binding to metal-responsive transcription factor-1, a phenomenon reflecting the fact that MT also provides protection from intracellular zinc excess [198,199,200]. A number of case-control studies have found that serum zinc levels are lower in PD patients than in matched healthy controls, as confirmed by meta-analysis [201,202]. The impact of zinc supplementation on risk for or control of PD, in rodents or humans, does not appear to have been assessed. 

Another natural scavenger of peroxynitrite-derived radicals is uric acid. Considerable prospective epidemiology reveals that, at least in men, relatively high serum urate is associated with reduced risk for, and slower progression of, PD [203,204,205,206]. The failure of some studies to confirm this phenomenon for women may reflect the fact that their serum urate levels tend to be lower than those of men, and hence may not be high enough to provide important protection. In seeming paradox, gout is not associated with a lower risk for PD, and in some studies is associated with increased risk [207,208]. This phenomenon remains unexplained; a credible possibility is that inflammation associated with gout tends to increase PD risk. In other words, hyperuricemia may reduce PD risk unless and until it precipitates gout—then the resulting systemic inflammation tends to promote brain inflammation and drive up PD risk. In any case, hyperuricemia, unless it precipitates gout, appears to lessen PD risk. 

The fact that high intakes of dairy products are associated with increased PD risk in men, has been rationalized by the observation that milk protein exerts a uricosuric effect [209,210]. Although evaluation of elevated plasma urate in rodent models of PD has proved difficult owing to ample expression of uricase in these animals, a recent study in which uricase was inhibited with potassium oxanate showed that concurrent dietary administration of uric acid provided protection from 6-hydroxydopamine-induced PD [211]. In humans, serum urate levels can be increased with reasonable safety by supplementation with its biosynthetic precursor inosine [212,213,214]. In pilot studies in PD patients, individually tailored dosing schedules (500–3000 mg daily) have been shown to maintain serum urate in the range of 6–8 mg/dl [215,216]. Episodes of urolithiasis were seen in a significant minority of patients, but could be managed with dosage reduction. Controlled studies evaluating the impact of supplemental inosine on clinical progression in PD patients can be expected. This strategy presumably could not be recommended for use in primary prevention, as it can increase risk for urate urolithiasis and requires clinical supervision, but hopefully will prove useful in PD therapy.

The xanthophyll carotenoid astaxanthin appears to be one of the most effective oxidant scavengers for bilipid layer membranes, such as the mitochondrial inner membrane that hosts the respiratory chain. Notably, astaxanthin protects membranes from peroxynitrite-derived radicals [217]. Oxidant damage to the respiratory chain may play an important role in initiating and driving PD. Astaxanthin has been found to be more protective for mitochondrial membranes than alpha-tocopherol [218,219]. Rodent and cell culture studies with astaxanthin show that it provides protection from the mitochondrial toxins typically used to induce a PD syndrome in rodents [220,221,222,223,224]. Hence, it may be reasonable to include supplemental astaxanthin in a regimen intended for PD prevention and control. 

## 6. Keeping Mitochondria Efficient by Optimizing Mitophagy—H_2_S and Plant-Based Diet

Dopaminergic neurons in the SN are unusually sensitive to the energy deficit associated with mitochondrial dysfunction, as their highly arborized axons stretch over 4 meters and make over a million synaptic connections [225]. Hence, efficient mitochondrial ATP generation is crucial to the proper function and survival of these neurons. It is not accidental that many of the chemical agents used to induce Parkinsonian pathology in rodents, such as 6-hydroxydopamine, MPTP, and rotenone, are mitochondrial toxins. Analogously, epidemiological studies suggest that exposure to pesticides potentially toxic to mitochondria may play a triggering role in the induction of PD [226,227,228]. Mitochondrial dysfunction associated with reduced complex I activity in at-risk dopaminergic neurons is a characteristic feature of PD [229,230]. Protection of mitochondria through dietary or nutraceutical measures may have important potential for preventing the initiation of the vicious cycle of inflammation and neuronal death that drives the progression of PD. 

The E3 ubiquitin ligase Parkin plays a key role in maintenance of efficient mitochondrial function. Parkin promotes mitophagy of depolarized mitochondria, while also aiding mitochondrial biogenesis [231,232]. The protein PINK1, after binding to the outer membrane of depolarized (hence damaged and inefficient) mitochondria, recruits Parkin; Parkin’s ubiquitin ligase activity promotes proteasomal degradation of certain proteins on the outer mitochondrial membrane which tether it to other structures—a prerequisite for mitophagy—and also attaches ubiquitin chains to the outer membrane, marking it for inclusion in developing autophagosomes [231]. People who are homozygous for loss-of-function mutations in Parkin develop early-onset PD [233,234]. Moreover, homozygosity for loss-of-function mutations in other proteins required for efficient mitophagy—such as PINK1 and LRRK2—likewise predisposes to early PD [235]. Hence, accumulation of damaged mitochondria may be a “spark” that can ignite the neuroinflammation underlying PD.

Overexpression of Parkin in the SN via lentiviral vectors protects mice from PD induced with 6-hydroxy-dopamine, MTPT, or mutant ASYN [236,237,238]. This suggests that more practical measures which boost Parkin expression or activity in striatal dopaminergic neurons may confer protection from PD. Sulfhydration of Parkin has been reported to boost its activity, whereas S-nitrosylation inhibits it [10,11,239]. Hence, measures which promote H_2_S production (taurine, NAC) may support Parkin activity both by direct sulfhydration, and by suppressing nitrosylation of this protein by down-regulating iNOS induction. Sulfhydration of Parkin is notably diminished in the striatum of PD patients [239]. 

Parkin expression is driven at the transcriptional level by the transcription factor ATF4 [240,241]. Certain stress conditions which activate eIF2alpha kinase are known to selectively increase translation of ATF4 mRNA, boosting the protein expression of ATF4 [242]. One of the stressors which activates eIF2alpha kinase is essential amino acid depletion, which is detected by the kinase GCN2; when specific essential amino acids are in short supply, their uncharged transfer RNAs can bind to GCN2, activating its kinase activity, which in turn very selectively phosphorylates and activates eIF2alpha kinase [243,244,245]. 

Amino acid status regulates not only GCN2 activity, but also that of the crucial regulatory kinase mammalian target of rapamycin complex 1 (mTORC1). Increased cellular levels of leucine, arginine, and of the methionine metabolite S-adenosylmethionine boost mTORC1 activity by suppressing mechanisms which turn off this activity [246,247,248,249,250,251,252]. Crucially, mTORC1 acts to inhibit expression of PINK1 at the transcriptional level; this effect might be mediated, in part, by transcriptional repression of FOXO1a, which binds to the promoter of the PINK1 gene and promotes its transcription [253,254,255]. Since PINK1 is required for recruitment of Parkin to damaged depolarized mitochondria, mTORC1 functions to suppress mitophagy [253,254,255]. Conversely, low-protein diets can be expected to boost mitophagy by concurrent activation of GCN2 and de-activation of mTORC1. As might be expected, certain genetic variants of PINK1 have been linked to autosomal recessive early-onset PD [256]. 

Plant-based diets of modest protein content tend to be rather low in certain essential amino acids—notably methionine and lysine—both because their total protein content is modest, and because plant protein tends to be relatively low in these particular amino acids [257]. Hence, there is reason to suspect that diets of this type may moderately up-regulate GCN2 activation [223] while down-regulating mTORC1 activity [258]. Indeed, up-regulation of GCN2 may explain why plasma IGF-I levels tend to be moderately lower in vegans than in vegetarians or omnivores [258]. Hepatic GCN2 activation is at least partially responsible for the marked elevation of plasma fibroblast growth factor-21 (FGF21) observed in rats placed on a low-protein diet [259]. Increased levels of plasma FGF21 are likewise seen when humans are put on an 8% protein diet, and have been found to be markedly higher in long-term vegans than in omnivores [260,261]. (It is notable that the traditional quasi-vegan diet of Okinawans, which was subsequently associated with the world’s highest proportion of centenarians, provided about 9% protein, mostly of plant origin [262]). 

During the last century, age-adjusted prevalence of PD, assessed by door-to-door community visits using standardized diagnostic measures, was found to be roughly a fifth as high in Nigerians or rural Chinese as in Americans, including blacks in Mississippi [263,264,265]. Although differential mortality following PD onset might explain a part of this difference, it seems reasonable to conclude that environmental factors were responsible for the bulk of it. Other epidemiology likewise suggests that Africans and East Asians were at decidedly lower risk for PD at a time when these groups were predominantly practicing quasi-vegan diets of modest protein content [266]. The two largest prevalence studies conducted to date focused on China, and found that PD was roughly 3 times more common in cities than in rural areas—concordant with greater consumption of animal products in cities [267,268]. 

Some years ago, these considerations led this author to propose that plant-based diets might aid PD prevention [266]. Although low-protein diets are known to aid levodopa control of PD symptoms by evening out variations in brain levodopa levels (a post-prandial surge in plasma branched-chain amino acids following a protein-rich meal competitively inhibits brain levodopa uptake), what is proposed here is that such diets might aid prevention or slow progression of PD by boosting striatal Parkin levels [269,270,271]. Although the impact of plant-based diets on rate of PD progression has not been assessed clinically, a few anecdotal reports, published or online, are consistent with benefit in this regard [272,273]. The impact of low-protein diets on striatal Parkin/PINK1 expression and progression of neuronal loss could readily be studied in rodent PD models. Whole-food plant-based diets might also afford some protection from PD as they are high in phytochemicals—some of which are antioxidant phase 2 inducers, such as ferulic acid—and are relatively low in fat-soluble neurotoxic contaminants (which accumulate in animal fat) linked to increased PD risk [274,275,276,277]. Increased dietary flavonoid intakes have been linked to decreased PD risk [278,279].

A dietary factor with potential for up-regulating mitophagy is the polyamine spermidine, which can inhibit an acetyltransferase, EP300, which functions to suppress both autophagy and mitophagy [280,281]. Dietary spermidine content was shown to correlate inversely with total mortality and cancer-related mortality in prospective epidemiological studies, possibly pointing to an “anti-aging” role for up-regulated autophagy/mitophagy [281,282,283]. Levels of spermidine in the cerebrospinal fluid of PD patients were found to be lower than those of controls, and dietary supplementation with this compound is protective in rotenone-treated rats [284,285]. In light of the fact that corn appears to be the richest dietary source of spermidine that is commonly consumed, it is intriguing that, two decades ago, Japanese scientists reported that age-adjusted mortality from PD in the provinces of Japan correlated inversely with the extent to which corn was grown in those provinces—prompting the speculation that “corn may prevent Parkinson’s disease” [286,287,288]. Spermidine is not yet available as a nutraceutical, so those who want a spermidine-rich diet are best advised to eat corn regularly; further studies should be employed to examine whether the residue from the manufacture of corn syrup could be employed to produce a spermidine supplement.

## 7. Controlling Intracellular Calcium with Calcium Channel-Blocker Drugs—and Magnesium? 

Curiously, users of dihydropyridine calcium channel-blocker drugs—which target the L-type voltage-sensitive Cav1.3 channels expressed by dopaminergic neurons of the SN—are at decreased risk for PD [289,290,291,292,293]. This might reflect the fact that, whereas healthy mitochondria can buffer the episodic flux of calcium through these channels—which function as rhythmically-activated pacemakers in dopaminergic neurons of the SN—maintaining a basal dopaminergic tone-impaired mitochondria does so less effectively, leaving neurons exposed to toxic intracellular calcium levels [294]. SN dopaminergic neurons with high expression of the calcium-buffering protein calbindin-D28k have a higher propensity to survive during progression of PD [295,296]. Calcium buffering by mitochondria can accelerate superoxide production by the electron transport chain—notably when complex I is partially inhibited—thereby amplifying the oxidative stress impacting dopaminergic neurons in PD and potentiating structural damage to mitochondria [297,298]. In addition, elevated intracellular calcium promotes ASYN aggregation in a way that is complementary to nitroxidative damage [299]. Despite their role as pacemakers in SN dopaminergic neurons, Cav1.3 channels can be inhibited without an important negative impact on function of these neurons, as channels for monovalent cations can “pinch hit” as pacemakers [300]. Hence, the inhibition of these channels, which is usually well tolerated systemically, is a very logical way to minimize exposure of SN dopaminergic neurons to intracellular calcium excess. Isradipine, a brain-penetrant dihydropyridine calcium channel blocker capable of inhibiting both Cav1.2 and Cav1.3 channels, has recently been tested in the STEADY-PD Phase III study in PD; unfortunately, no benefit was observed [301]. The authors have speculated that the dose schedule chosen—5 mg daily of immediate-release isradipine twice daily—may not have been high enough to achieve effective Cav1.3 inhibition. Indeed, a previous mouse study determined that tolerable clinical concentrations of isradipine capable of lowering blood pressure via inhibition of Cav1.2 smooth muscle channels are too low to achieve inhibition of Cav1.3 channels in the CNS [302]. This leaves unexplained the lower risk for PD in patients using calcium channel blockers—further investigation is required to explain whether isradipine was an unfortunate choice, or if the relationship was non-causal. 

Magnesium has been referred to as “nature’s physiologic calcium blocker” [303]. Its utility in this regard may reflect the ability of increased intracellular magnesium to compete with calcium for binding to certain EF-hand proteins such as calmodulin [304,305]. Whether improved magnesium status associated with increased intracellular magnesium might favorably influence the risk for, or the course of, PD—by counteracting adverse effects of calcium, or for other reasons—is a question that requires more study. Whereas calcium has a pro-aggregatory impact on ASYN, magnesium is reported to have an anti-aggregatory effect; how calcium and magnesium interact in this regard does not appear to have been studied [306]. Brain magnesium levels in various regions of the brain, including the basal ganglia, have been found to be lower in PD patients than in controls [307]. The only readily traceable case-control epidemiology correlating estimated dietary magnesium intakes with risk for PD found that those in the upper quartile of magnesium intake were at notably lower risk than those in the lower quartile: OR = 0.33 (95% CI: 0.13–0.81, p for trend = 0.007) [308]. Additional studies of this type are evidently needed. Increased extracellular magnesium is protective in cellular models of PD in which cell damage is induced by MPP+ or 6-hydroxydopamine [309,310]. Hence, although pertinent scientific studies are scarce, the possibility that magnesium might exert a protective role in PD is credible and deserves more attention from researchers.

## 8. Conclusions 

These considerations suggest that nutraceutical supplementation targeting the glial production and the neurotoxic oxidant activity of peroxynitrite may have considerable potential for the primary prevention of PD—and may have adjuvant utility in the clinical management of this disorder. Supplemental intake of spirulina or of PhyCB-enriched spirulina extracts may decrease glial generation of peroxynitrite by suppressing NAPDH oxidase-mediated production of its precursor superoxide. Glial induction of iNOS may be antagonized by supplementing with the anti-inflammatory antioxidant ferulic acid, supplementing with DHA-rich fish oil, maintaining good vitamin D status, supporting production of H_2_S with supplemental taurine and NAC, blocking adenosine receptors with caffeine, drinking green tea or supplementing with EGCG, and supporting ileal production of GLP-1 with butyrogenic dietary fiber. The toxic oxidant activity of peroxynitrite in dopaminergic neurons may be mitigated to some degree by boosting the expression of MT with zinc supplementation. Astaxanthin, which efficiently protects membranes from peroxynitrite, has important potential for preserving proper function of the mitochondrial respiratory chain. The nutraceuticals cited here are notable for their safety and their appropriateness for use in health-protective supplementation programs, and hence may be useful for primary prevention of PD. There is strong reason to suspect that, when administered in adequate doses and appropriate combinations, they could also be useful for slowing progression of clinical PD. Daily supplemental intakes of vitamin D in excess of 10,000 IU, or of zinc in excess of 100 mg, have potential for toxicity, and should be avoided. High supplemental intakes of zinc should be coupled with a modest supplemental intake of copper (1–2 mg daily) to prevent induction of copper deficiency [311]. There is little clinical experience with ferulic acid intakes exceeding one gram daily, so this is likewise not recommendable. 

Supplemental inosine, which can boost serum levels of the peroxynitrite scavenger uric acid, may also help to control progression of PD, though physician monitoring of dose will be necessary to minimize risk for urate crystal deposition. In light of evidence that dihydropyridine-type calcium channel blockers may slow PD progression—possibly by mitigating the pathogenic impact of mitochondrial impairment and diminishing ASYN aggregation—optimal intakes of “nature’s calcium antagonist”, magnesium, might also confer some protection. Table 1 provides suggestions regarding appropriate supplemental intakes of these agents. Figure 1 depicts how these agents may intervene in the pathogenesis of PD. The possibility that plant-based diets of modest protein content might aid prevention and control of PD by up-regulating striatal Parkin merits assessment; such diets are also good adjuvants to levodopa therapy. In any case, whole-food fiber-rich plant-based diets are recommendable as they can be expected to provide versatile health protection. Corn-rich diets may also up-regulate protective mitophagy via their spermidine content. Future prospects for prevention and control of PD may be bright.

## Figures and Tables

**Figure 1 ijms-21-03624-f001:**
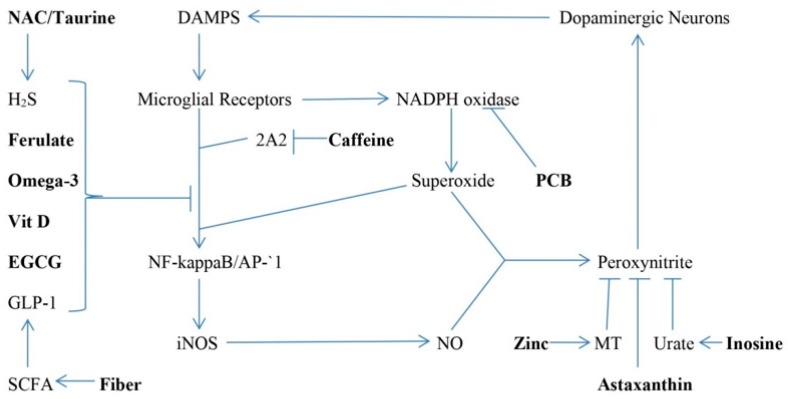
Vicious cycle of microglial activation and dopaminergic neuron death in pathogenesis of Parkinson’s disease (PD), depicting nutraceutical interventions that may provide protection.

**Table 1 ijms-21-03624-t001:** Nutraceuticals with potential for prevention/control of Parkinson’s disease. Suggested daily intake ranges.

Nutraceuticals	Suggested Daily Intake
PhyCB/Spirulina	100–200 mg/15–30 g
Ferulic Acid	500–1000 mg
DHA	1–2 g
Vitamin D	2000–10,000 IU
Green Tea Polyphenols	500–1000 mg
Taurine	2–6 g
N-Acetylcysteine	1200–1800 mg
Zinc	30–80 mg (plus 1–2 mg Cu *)
Astaxanthin	8–16 mg
Magnesium	200–400 mg
Inosine	500–3000 mg (physician supervised)

* To prevent zinc-induced copper deficiency.

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
