# Peer review of "Nutraceuticals Targeting Generation and Oxidant Activity of Peroxynitrite May Aid Prevention and Control of Parkinson’s Disease"

_ijms, 2020, doi:10.3390/ijms21103624_

Round 1

Reviewer 1 Report

The review article by Mark F. McCarty and Aaron Lerner has been written well, reads well, is engaging and informative. To improve the content of the manuscript, I have added some comments and suggestion in the annotated PDF of the original submission. I would also like to add those comments here for ease of reference. I have also suggested some minor punctuation and spelling changes in the annotated PDF (particularly, I would like the authors to hyphenate all the word components that act as a compound adjective or as an adjectival phrase modifying a noun; see Garner’s Modern English Usage, Fourth Edition for that.) My comments follow:

  1. Line 33: In the manuscript, please discuss how peroxynitrite compares with the role of the myeloperoxidase system in microglia and why is the former more notable (potentially than the other system)?
  2. Line 100: I would like you to comment on the fact that RAGE is generally a promiscuous receptor. Additionally, many other S100 proteins may be able to bind to it or activate it, especially the inflammation-associated ones like S100A8, S100A9, S100A12, along with S100B.
  3. Line 219: I would like the authors to consider the possibility that EGCG may be a pan-assay interfering compound (See PMIDs: 29202222, 28676405, 26313340, 28165734, 30422657, 28165734, 26900761 and 28074653). Please consider this possibility and discuss in that context, too. This is relevant with most other polyphenols. It would be interesting to discuss experiments to disprove that EGCG is a pan-assay interfering compound if such literature exists.
  4. Line 246: Please discuss the fact that there is no association between high uric acid (gout) levels and reduced incidence of PD. This literature does not agree with the discussion presented in this manuscript. An unbiased and more informative view is necessary. See PMID 30611222 and PMID 26330027.
  5. Line 342, please introduce isradipine in a line or so.
  6. Line 345: Please note that S100B and the three other inflammation-associated S100 proteins that I mentioned are EF-hand proteins. Is there any relevant literature regarding those proteins and magnesium in the context discussed in this manuscript? Please elaborate here as they seem relevant.

Thank you for the opportunity to review your manuscript and good luck. 

Author Response

Please note that all alterations to the manuscript have been highlighted in yellow.

Also note that, in addition to revisions addressing the concerns of the reviewers, we have added an additional paragraph under Keeping Mitochondria Efficient that discusses the ability of low-protein diets to boost mitophagy by down-regulating mTORC1 activity.  We became aware of these findings since submitting our manuscript. We have also added a short discussion regarding mechanisms of NADPH oxidase activation in the Targeting NADPH Oxidase section, a paragraph on astaxanthin under Scavenging Peroxynitrite-Derived Radicals, and a paragraph discussing spermidine and corn-rich diets to the Mitophagy section. 

Reviewer #1

Line 33: In the manuscript, please discuss how peroxynitrite compares with the role of the myeloperoxidase system in microglia and why is the former more notable (potentially than the other system)?

We now include, right before the Targeting NADPH Oxidase section, a paragraph briefly citing literature pertinent to the role of myeloperoxidase in PD.

Line 100: I would like you to comment on the fact that RAGE is generally a promiscuous receptor. Additionally, many other S100 proteins may be able to bind to it or activate it, especially the inflammation-associated ones like S100A8, S100A9, S100A12, along with S100B.

We have added a sentence mentioning this in the first paragraph under Blocking Induction of iNOS. 

Line 219: I would like the authors to consider the possibility that EGCG may be a pan-assay interfering compound (See PMIDs: 29202222, 28676405, 26313340, 28165734, 30422657, 28165734, 26900761 and 28074653). Please consider this possibility and discuss in that context, too. This is relevant with most other polyphenols. It would be interesting to discuss experiments to disprove that EGCG is a pan-assay interfering compound if such literature exists.

We apologize that we don’t understand this, or its significance in our discussion.  If this is important to the reviewer, could he/she clarify this issue for us?

Line 246: Please discuss the fact that there is no association between high uric acid (gout) levels and reduced incidence of PD. This literature does not agree with the discussion presented in this manuscript. An unbiased and more informative view is necessary. See PMID 30611222 and PMID 26330027.

Thank you for acquainting us with this remarkable and rather paradoxical finding.  We now discuss and interpret this in the section Scavenging Peroxynitrite-Derived Radicals.

Line 342, please introduce isradipine in a line or so.

Since we wrote our manuscript, the STEADY-PD Phase III study of israpidine has been completed with a null result.  We now discuss this, with more commentary on isradipine, in the paragraph under Controlling Intracellular Calcium.

Line 345: Please note that S100B and the three other inflammation-associated S100 proteins that I mentioned are EF-hand proteins. Is there any relevant literature regarding those proteins and magnesium in the context discussed in this manuscript? Please elaborate here as they seem relevant.

The pertinent role of S100B and related proteins in PD is as extracellular ligands for receptors such as RAGE.  We haven’t been able to find any literature indicating that increased extracellular magnesium might alter their ligand function through calcium competition. If this were so, it would be intriguing, as magnesium threonate could be employed to elevate extracellular magnesium levels in the brain.

We thank reviewer #1 for careful attention to our manuscript and insightful suggestions.

Reviewer 2 Report

Comments and Suggestions for Authors

The work is a complete and well-structured review of the common molecular mechanisms underlying Parkinson’s disease and the effects of different antioxidants.

  • In my opinion the title of the review is a little confusing, and may not come directly to the readers. I would change it more simple and immediate.

  • The paper would acquire importance if a figure - that summarizes the content -were inserted in the text In particular, a figure should describe the main mechanisms of the pathology. Another figure should summarize the protective mechanisms of the indicated nutraceuticals. In this way it would be easier for readers to interpret review.

  • Some nutraceuticals you speak of have toxic effects when used in high concentrations. You say correctly in the conclusions that these componds have protective effects “when administered in adequate doses and appropriate combinations”. Nevertheless possible toxical effects should be mentioned.

Author Response

To reviewer No 2

Please note that all alterations to the manuscript have been highlighted in yellow.

Also note that, in addition to revisions addressing the concerns of the reviewers, we have added an additional paragraph under Keeping Mitochondria Efficient that discusses the ability of low-protein diets to boost mitophagy by down-regulating mTORC1 activity.  We became aware of these findings since submitting our manuscript. We have also added a short discussion regarding mechanisms of NADPH oxidase activation in the Targeting NADPH Oxidase section, a paragraph on astaxanthin under Scavenging Peroxynitrite-Derived Radicals, and a paragraph discussing spermidine and corn-rich diets to the Mitophagy section. 

  1. In my opinion the title of the review is a little confusing, and may not come directly to the readers. I would change it simpler and more immediate.

We have shortened the title a bit, and think that it should be reasonably clear to most readers.

  1. The paper would acquire importance if a figure - that summarizes the content -were inserted in the text In particular, a figure should describe the main mechanisms of the pathology. Another figure should summarize the protective mechanisms of the indicated nutraceuticals. In this way it would be easier for readers to interpret review.

We have produced a figure which depicts the main mechanisms of pathology, and shows where the discussed nutraceuticals may intervene in them.  We agree that this should aid comprehensibility of our paper. 

  1. Some nutraceuticals you speak of have toxic effects when used in high concentrations. You say correctly in the conclusions that these componds have protective effects “when administered in adequate doses and appropriate combinations”. Nevertheless possible toxical effects should be mentioned.

In the first paragraph under Summing Up, we have added sentences cautioning against excessive intakes of vitamin D, zinc, and ferulic acid.

Thank you for your useful suggestions and appreciative comments.  

Round 2

Reviewer 2 Report

in my opinion the work has been improved and is ready for publication